# Role of NRF2 in Ovarian Cancer

**DOI:** 10.3390/antiox11040663

**Published:** 2022-03-30

**Authors:** Giovanni Tossetta, Sonia Fantone, Eva Montanari, Daniela Marzioni, Gaia Goteri

**Affiliations:** 1Department of Experimental and Clinical Medicine, Università Politecnica delle Marche, 60126 Ancona, Italy; s.fantone@pm.univpm.it (S.F.); d.marzioni@univpm.it (D.M.); 2Clinic of Obstetrics and Gynaecology, Department of Clinical Sciences, Salesi Hospital, Azienda Ospedaliero Universitaria, 60126 Ancona, Italy; 3Section of Legal Medicine, Azienda Ospedaliero Universitaria Ospedali Riuniti, 60126 Ancona, Italy; eva.montanari@ospedaliriuniti.marche.it; 4Department of Biomedical Sciences and Public Health, Università Politecnica delle Marche, 60126 Ancona, Italy; g.goteri@staff.univpm.it

**Keywords:** NRF2, cisplatin, chemotherapy, ovarian cancer, ovarian preservation

## Abstract

Among gynaecologic malignancies, ovarian cancer is one of the most dangerous, with a high fatality rate and relapse due to the occurrence of chemoresistance. Many researchers demonstrated that oxidative stress is involved in tumour occurrence, growth and development. Nuclear factor erythroid 2-related factor 2 (NRF2) is an important transcription factor, playing an important role in protecting against oxidative damage. Increased levels of Reactive Oxygen Species (ROS) activate NRF2 signalling, inducing the expression of antioxidant enzymes, such as haem oxygenase (HO-1), catalase (CAT), glutathione peroxidase (GPx) and superoxide dismutase (SOD), that protect cells against oxidative stress. However, NRF2 activation in cancer cells is responsible for the development of chemoresistance, inactivating drug-mediated oxidative stress that normally leads to cancer cells’ death. In this review, we report evidence from the literature describing the effect of NRF2 on ovarian cancer, with a focus on its function in drug resistance, NRF2 natural and synthetic modulators and its protective function in normal ovarian preservation.

## 1. Introduction

Among gynaecologic malignancies, ovarian cancer is a leading cause of death because it is often diagnosed at an advanced stage of disease. Although the majority of ovarian cancers are epithelial ovarian cancer (EOC), different histological subtypes of EOC have been identified, including serous (the most common), endometrioid, clear cell and mucinous cancers. The current treatment for advanced ovarian cancer is surgery, followed by platinum/taxane chemotherapy, but many patients relapse within 18 months due to chemoresistance onset. This explains the high mortality rates of this type of cancer [1].

Oxidative stress can affect all phases of oncogenesis, i.e., cancer initiation, promotion, and progression, activating many transcription factors, such as nuclear factor (NF)-κB, peroxisome proliferator-activated receptor γ (PPARγ), p53, hypoxia inducible factor 1α (HIF-1α) and Nuclear Factor Erythroid 2-Related Factor 2 (NFE2L2 or NRF2). These factors can induce the expression of numerous genes involved in many cellular processes, including inflammatory responses, apoptosis, cell proliferation and differentiation [2].

Cell metabolism, infection, exposure to carcinogens and environmental toxicants are the main producers of endogenous and exogenous Reactive Oxygen Species (ROS), highly reactive molecules, such as hydroxyl radical (OH^−^), hydrogen peroxide (H_2_O_2_) and superoxide (O_2_^−^), as well as reactive nitrogen species that can damage cellular DNA, proteins and lipids, if not eliminated [3,4,5]. Therefore, cells developed different mechanisms to tolerate ROS presence. In fact, cells can tolerate low ROS levels acting as second messengers and modulating different intracellular pathways involved in many cellular processes, including cell proliferation, differentiation and migration [3,6]. However, high levels of ROS cause cell death because of cellular components oxidation as membrane lipids, proteins and DNA [3]. ROS can be neutralized by glutathione (GSH), coenzyme Q, lipoic acid or antioxidant enzymes, such as superoxide dismutase (SOD), catalases (CATs), thioredoxins (Trxs), peroxiredoxins (Prxs), reductases and peroxidases [3,7].

Moreover, ROS scavenging enzymes, carrying antioxidant response elements (AREs) in their promoter regions [8], represent an intrinsic defence mechanism to avoid cell damage caused by ROS. A key regulator of AREs is NRF2, a basic leucine zipper transcription factor that binds ARE regions present in the promoter of many antioxidant enzyme-activating genes [8].

Normally, NRF2 is directly bound to a negative regulator Kelch-like ECH-Associated Protein 1 (KEAP1) that is further bound to Cullin 3 (CUL3) and RING-box protein 1 (RBX1)/E3-ubiquitin ligase, forming the KEAP1/CUL3/RBX1 E3-ubiquitin ligase complex that targets NRF2 for proteasomal degradation. In the presence of oxidant stimuli, the binding between ROS and cysteine residues of KEAP1 leads to a conformation change in KEAP1. The latter causes the inhibition of NRF2 ubiquitination and its translocation into the nucleus (active form), with consequent binding to the ARE regions in the promoter of antioxidant genes, inducing their expression (Figure 1) [9]. The NRF2 pathway also plays a key role in carcinogenesis because it inhibits apoptosis, promoting cell proliferation and chemoresistance. Thus, the NRF2 pathway is emerging as a chemotherapeutic target in many types of cancer [10].

To date, platinum drugs (cisplatin, carboplatin, and oxaliplatin), alone or in combination with other drugs, are the most used clinical agents in chemotherapy against ovarian cancer, and among them, cisplatin (cis-diamminedichloroplatinum II, CDDP) is one of the most efficient. Cisplatin forms both mono-adducts, covalently interacting with N7-guanine in DNA, and intra- and/or inter-strand crosslinks [11,12,13]. Cell apoptosis, due to the block of DNA synthesis and transcription, is the consequence of these alterations due to cisplatin [12,14]. Unfortunately, cisplatin resistance is one of the major problems occurring in cisplatin-based chemotherapy. Interestingly, it has been shown that the cisplatin-resistant human ovarian cancer SKOV3 cell line, which retains high levels of GSH, could be sensitized to cisplatin treatment by inhibiting NRF2. In fact, NRF2 inhibition led to GSH depletion, increasing cisplatin cytotoxicity and proving that the NRF2 pathway plays a key role in cisplatin resistance [15]. Interestingly, this cell line has high basal ROS levels that, enabling NRF2 nuclear translocation, favour NRF2 antioxidant activity and allow a better cell resistance to endogenous oxidant agents. Additionally, NRF2 and KEAP1 basal levels are cell line specific and positively correlate both to cell growth rates and basal ROS, in the following ovarian cancer cell lines: PEO1, PEO4, PEO6, SKOV3, OVCAR3 and OVCAR4. In particular, basal ROS levels and cell growth are higher in SKOV3, OVCAR3 and PEO1 compared to the other cell lines. The high basal ROS levels favour NRF2 stability and antioxidant activity, allowing a better cell resistance to the exogenous oxidant agents [16]. Similar findings were found in Doxorubicin (DR) resistance, an oxidative chemotherapeutic drug used in the treatment of many solid tumours, including ovarian cancer [17,18].

NRF2/ARE binding and NRF2 target genes expression are lower in the A2780 cell line, which is highly sensitive to DR, compared to resistant ovarian carcinoma SKOV3 and OV90 cell lines. Contrarily, doxorubicin-resistant A2780DR cells show increased NRF2 activity, enhanced GSH1 and GSH contents, suggesting that NRF2 might be a key factor in DR sensitivity [19]. Thus, an adaptive activation of the NRF2 pathway may participate in the development of DR-acquired resistance. NRF2 silencing in OV90 cells lead to a reduction in GSH levels and retard cell growth sensibilizing OV90 cells to DR treatment. This causes increased apoptosis, suggesting NRF2 as a molecular target to restore DR sensitivity and repress tumour growth [20].

E26 transformation-specific (ETS) proteins are a family of 28 transcription factors, containing a highly conserved ETS domain for DNA binding [21]. They are considered important onco-drivers and play a pivotal role in the progression of many types of cancer [21,22,23]. Interestingly, H_2_O_2_ upregulates Ets-1, a member of ETS proteins family, expression in both OV2008 and C13 ovarian carcinoma cell lines. Moreover, H_2_O_2_ treatment increases Ets-1 expression by NRF2 binding to ARE in the Ets-1 promoter, suggesting that Ets-1 is clearly modulated by ROS in cancer cells via NRF2 signalling [24].

The aim of this review is to provide an overview of the current literature, regarding the role of NRF2 in ovarian cancer and normal ovarian preservation, with a focus on its cellular modulators and targets.

## 2. NRF2 in Ovarian Cancer Tissues

Nicotinamide adenine dinucleotide phosphate (NAD-P) H/quinone oxidoreductase 1 (NQO1) is a metabolizing enzyme, capable of generating antioxidant forms of ubiquinone and vitamin E after free radical exposure. It has an important role in cellular defence mechanisms against oxidative stress, acting as antioxidant enzyme. Interestingly, both NQO1 and NRF2 are highly expressed in ovarian carcinoma compared to normal and precancerous lesions, showing a positive correlation in the different lesions. In addition, NRF2 is expressed in both the nucleus and cytoplasm of ovarian cells (90 ovarian carcinomas of different grades and 10 normal ovarian tissues) and it significantly increases as ovarian carcinoma stage advances [25]. Therefore, the authors suggest that NQO1 and NRF2 may be considered therapeutic targets for ovarian cancer care and possible early diagnostic biomarkers. Similarly, Liew and colleagues reported that serous carcinoma has higher KEAP1 cytoplasmic, NRF2 nuclear and lower E-cadherin membrane positivity than mucinous, endometrioid and clear cell cancers, studying 108 cases (47 serous, 23 mucinous, 13 endometrioid and 25 clear cell). Moreover, KEAP1 expression is further increased in serous carcinoma from elderly patients. Multivariate analysis identified International Federation of Gynecology and Obstetrics (FIGO) staging and NRF2 expression as prognostic factors. Thus, NRF2 and Keap1 can be considered key players in serous carcinoma [26].

These results are partially in agreement with another study on NRF2 and KEAP1 evaluation as prognostic indicators. In particular, low KEAP1 expression is associated with both disease recurrence and death, while high KEAP1 expression is predictive of better overall and disease-free survival, suggesting KEAP1 as an independent prognostic factor, not linked to NRF2. Contrarily, high NRF2 expression shows a better overall and disease-free survival, but the results are not statistically significant, and no significant association is detected among chemoresistance, NRF2 and KEAP1 expression [27].

Others reported that nuclear NRF2 expression is low in serous, clear cell, and endometrioid ovarian carcinomas, while it is high in mucinous subtype. Moreover, low nuclear NRF2 expression in those carcinomas is associated with aging. No significant difference was detected in oestrogen receptor α (ERα) expression, comparing all ovarian cancer subtypes, but low-grade carcinomas showed a significantly higher ERα expression compared to high-grade ones. Interestingly, NRF2 cytoplasmic expression was positively correlated with ERα expression in serous ovarian carcinoma and their expressions have been associated with longer overall survival. This suggests that the inhibition of NRF2 may represent an effective therapeutic strategy for OEC treatment [28]. In addition, cytoplasmic NRF2 expression is significantly correlated with both progesterone receptor A and B (PRA and PRB) expressions and associated with increased overall survival [29].

Looking at the studies discussed in this section (and summarized in Table 1), it is possible to see that many of them reach contradictory conclusions. This can be due to the small cohorts’ study and to the limitations of the technique (immunohistochemistry).

## 3. NRF2 Cellular Modulators in Ovarian Cancer

MicroRNAs have been associated with many pathological processes, including human pregnancy complication [30,31,32], cancer progression [33,34,35,36] and chemo- or radiotherapeutic resistance [37,38]. MiR-181d is highly expressed in ovarian tissues of DDP-(cisplatin) resistant patients and in the cisplatin-resistant A2780/DDP cell line. Moreover, ectopic expression of miR-181d increases DDP resistance in A2780 non-resistant cells. miR-181d negatively regulates O-linked N-acetylglucosamine (GlcNAc) transferase (OGT) expression by targeting its mRNA in 3′UTR. OGT is an essential enzyme for KEAP1 glycosylation and stabilization that implies NRF2 inactivation [39], resulting in increased DDP resistance in an A2780 cell model [40]. This study highlighted a key role of miR-181d in modulating DDP resistance in ovarian cancer through the OGT/KEAP1/NRF2 axis.

In addition to microRNAs, there are an increasing number of studies showing an important role of long non-coding RNAs (lncRNAs) in carcinogenesis [41,42,43]. In particular, LncRNA H19 is encoded by the H19 gene and is one of the first discovered lncRNAs [44]. LncRNA H19 plays a pivotal role in tumorigenesis and malignant progression, promoting many cell processes, including cell growth, invasion, migration and epithelial-mesenchymal transition [45]. Interestingly, lncRNA H19 levels are significantly increased in cisplatin-resistant A2780/CDDP cells. Moreover, H19 levels are higher in patients with high-grade serous ovarian cancer (HGSC) and correlate with cancer recurrence. Furthermore, A2780/CDDP cells with lncRNA H19 knockdown result in the recovery of cisplatin sensitivity and reduce the expression of six NRF2-modulated proteins, such as NQO1, Glutathione-Disulfide Reductase (GSR), Glucose-6-phosphate Dehydrogenase (G6PD), Glutamate-Cysteine Ligase catalytic subunit (GCLC), Glutamate-cysteine ligase regulatory subunit (GCLM) and Glutathione S-Transferase Pi 1 (GSTP1), involved in oxidative stress control. Additionally, these cells have low glutathione levels and are significantly more sensitive to H_2_O_2_ treatment, suggesting that lncRNA H19 may play a key role in cisplatin-resistance-modulating NRF2 signalling [46]. Moreover, mutations of common oncogenes, such as KRAS, BRAF, and MYC, increase NRF2 transcription and activity in malignant cells, protecting tumour cells from ROS cytotoxic effects induced by chemotherapeutic drugs, such as cisplatin, and play a key role in cisplatin resistance [47,48].

Protein p62/SQSTM1 (sequestosome 1) is an antiapoptotic mediator acting as a cargo protein that identifies (via a ubiquitin-binding domain) and delivers specific organelles and protein aggregates to autophagosomes for degradation (a process called selective autophagy) [49]. In this way, autophagy can allow tumour cell survival and growth, facilitating the supply of metabolic demands during tumour progression [50,51]. Autophagy is also involved in processes inducing cisplatin resistance of human ovarian cancer cells [52], and it is higher in cisplatin-resistant SKOV3/CDDP cells than cisplatin-sensitive SKOV3 cells. In particular, p62 interacts with the KEAP1/NRF2/ARE pathway i.e., after p62 phosphorylation, it physically interacts with KEAP1 and, subsequently, NRF2 transfers to the nucleus, activating the expression of antioxidant genes in SKOV3/DDP cells. Therefore, p62 can prevent ROS-induced apoptosis activating the KEAP1/NRF2/ARE signalling [53].

Another protein, Sirtuin 5 (SIRT5), belonging to the Sirtuin family of proteins (Sirt1–7), is involved in the regulation of multiple cellular processes, including glycolysis, fatty acid oxidation, nitrogen metabolism and drug resistance in cancer cells [54,55]. Interestingly, SIRT5, a mitochondrial NAD-dependent deacetylase, is increased in ovarian cancer tissues, predicting a poor response to chemotherapy. Moreover, SIRT5 levels are higher in cisplatin-resistant SKOV-3 and CAOV-3 ovarian cancer cells than in cisplatin-sensitive A2780 cells. SIRT5 overexpression facilitates ovarian cancer cell growth and cisplatin-resistance in an in vitro A2780 cell model, because SIRT5 suppresses cisplatin-induced DNA damage by increasing NRF2 and Haem Oxygenase 1 (HO-1) expression [56].

The KEAP1/CUL3/RBX1 E3-ubiquitin ligase complex is a key inhibitor of NRF2 levels [57]. It is known that the NRF2 pathway is a critical starting pathway for oxidative stress response and this pathway is constitutively active in serous ovarian carcinomas (OVCA) [58,59]. In fact, it has been reported that almost 90% of OVCA cases exhibit loss-of-function alterations in any components of the above inhibitory complex. Copy-number loss (CNL) is the most prominent disruption mechanism and most frequently observed in the RBX1 component. Consequent reduced mRNA complex expression enhances NRF2 target gene expression, suggesting that a remarkably high frequency of DNA and mRNA alterations of the KEAP1/CUL3/RBX1 complex leads to high levels of NRF2, found in OVCA [60].

The p53 upregulated modulator of apoptosis (PUMA) is a member of the Bcl-2 family, localized in the mitochondria and involved in mitochondrial-dysfunction-mediated apoptosis [61]. PUMA is an antagonist of both BCL-XL and MCL-1 antiapoptotic Bcl-2 family members, then acting as a proapoptotic factor [62]. Although the mechanism of action of PUMA remains unclear, it is possible that it could play a key role in inhibiting tumour growth [63]. PUMA, mainly located into the mitochondria, induces apoptosis by ROS production and increases both NRF2 and HO-1 expression in transfected A2780 and SKOV3 cells. Thus, PUMA induces ROS generation, damaging DNA and leading to cell apoptosis, but it enhances NRF2/HO-1 expression, providing an antioxidant response to ROS-mediated oxidative stress [64].

The same research group showed NRF2 nuclear (active form) localization in the majority of EOC specimens analysed with more frequency in clear cell EOC subtype and an upregulation of NRF2 target genes. In fact, 29% of clear cell carcinoma samples shows genetic mutations of the KEAP1 sequence. Importantly, patients with an active NRF2 pathway show resistance to platinum-based therapy and lower overall survival compared with those patients where the NRF2 pathway is inactivated [58]. Others showed that KEAP1 mutations have a key role in NRF2 activation and in platinum-based therapy resistance onset in EOC. In this study, the heterodimer, between the wild-type KEAP1 and the mutant KEAP1 subunits, is inactive and is unable to repress NRF2 in tumours with KEAP1 mutations [65]. Pylväs-Eerola and colleagues showed an increased expression of KEAP1 after Platinum-based neoadjuvant chemotherapy, suggesting a role of KEAP1 in degrading NRF2, promoting chemotherapy response [66]. Studies discussed in this chapter has been summarized in Table 2. 

## 4. NRF2 Cellular Targets in Ovarian Cancer

Oestrogen Receptor α (ERα) mediates the effects of female steroid hormones and can be considered a key regulator of apoptosis and cell proliferation in EOC [67,68,69]. In addition, Progesterone Receptor (PGR) expression has been associated with improved overall survival (OS) and progression-free survival (PFS), probably due to its anti-proliferative effect [70,71]. Interestingly, ERα is reduced in all ovarian cancer cell lines (OVCAR3, ES2, UWB1.289, and TOV112D) compared to the benign cell line HOSEpiC. Contrarily, NRF2 is highly expressed in ovarian cancer cell lines compared with the benign HOSEpiC cell line. Interestingly, NRF2 silencing induces an increase in ERα and PGR mRNA expressions in OVCAR3 cell lines [28,29], confirming a role of NRF2 in regulating ERα and PGR expression in ovarian cancer cell lines.

Another molecule involved in many cellular processes, including apoptosis, cell proliferation and differentiation, is CD99, a transmembrane glycoprotein coded by the MIC2 (MHC class I related antigen 2) gene [72] that is considered a prognostic marker of ovarian cancer [73]. Wu and colleagues demonstrated that CD99 is highly expressed in cisplatin-resistant ovarian cancer cells, using in vitro models (A2780/CDDP and COC1/CDDP) and ovarian tissues. Contrarily, CD99 is poorly expressed in cisplatin-sensitive ovarian cancer cells (A2780 and COC1) and ovarian tissues. Thus, CD99 overexpression results in cisplatin resistance, while CD99 knockdown sensitizes ovarian cancer cells to cisplatin. In addition, NRF2 overexpression increases CD99 expression and cell viability after cisplatin treatment in cisplatin-sensitive cells. Conversely, NRF2 knockdown decreases CD99 expression and cell viability after cisplatin treatment in cisplatin-resistant cells. We can summarize that simultaneous CD99 overexpression and reactivated cisplatin resistance in ovarian cancer cells suggest that CD99 can be an NRF2 downstream gene and that NRF2 can modulate cisplatin resistance by CD99 up or downregulation [74].

Another study demonstrated that NRF2 silencing represses NRF2 signalling in SKOV3 cells and causes cell growth G0/G1 phase arrest. Moreover, NRF2 silencing induces tumour growth retardation in mouse xenografts and a significant decrease in ErbB2 expression, a member of the human epidermal growth factor (EGF) receptor family that plays an essential role in cell proliferation and differentiation [75]. At the same time, ErbB2 downregulation leads to a pAKT decrease and p27 increase, enhancing the effect of NRF2 knockdown in SKOV3 growth [76]. Furthermore, NRF2 inhibition increases the sensitivity to docetaxel cytotoxicity and apoptosis, providing important findings on its role in docetaxel-based chemotherapy.

Aldo-keto reductases comprise AKR1C1–AKR1C4, four enzymes that catalyse NADPH-dependent reductions [77] but are also involved in chemoresistance [78]. Interestingly, NRF2 knockdown leads to decreased AKR1C1, AKR1C2 and AKR1C3 expression and increased ROS production after cisplatin treatment in SKOV3 cells. Moreover, a significant activation of the pJNK/p38 pathway and decreased phosphorylation of Activating transcription factor-2 (ATF2), a member of the leucine zipper family of DNA-binding proteins, implicated as a tumour suppressor, was observed in NRF2 knockdown cells, suggesting that NRF2 markedly modulates cisplatin resistance, regulating the AKR family members via the activation of the pJNK/p38 pathway [79].

Hepatocyte Growth Factor Receptor (HGFR/c-MET) and Epidermal Growth Factor Receptor (EGFR) are cell surface receptor tyrosine kinases (RTK), primarily expressed by epithelial cells that modulate different cell functions, including cell proliferation, survival and motility. Additionally, they can induce chemotherapy resistance [80]. Interestingly, NRF2 silencing increases miR-206 expression and reduces c-MET and EGFR levels through the direct binding to the 3′-untranslated region of the c-MET and EGFR genes in SKOV3 cells. The increased miR-206 levels have repressed c-MET/EGFR expression, inhibiting cell proliferation. In addition, miR-206 inhibits BCRP/ABCG2 expression, the human breast cancer resistance protein (BCRP, gene symbol ABCG2) that is an ATP-binding cassette (ABC) efflux transporter, and increases doxorubicin sensitivity, suggesting NRF2 as a modulator of BCRP/ABCG2 expression via miR-206 regulation [81].

ABC transporters are a superfamily of thirteen transporter (ABCA3, ABCB1 (MDR1), ABCB6, ABCB8, ABCB10, ABCB11, ABCC1 (MRP1), ABCC4, ABCC9, ABCD3, ABCD4, ABCE1, and ABCF2) proteins that transport various molecules across membranes, utilizing ATP as an energy source [82]. It has been reported that many of the ABC transporters are involved in chemoresistance occurrence, including ABCF2 that is involved in cisplatin resistance [83]. Interestingly, it has been reported that ABCF2 has a functional antioxidant response element (ARE) in its promoter region, suggesting ABCF2 as an NRF2 target gene. In fact, A2780 cells, overexpressing NRF2, contain high levels of ABCF2 and are more resistant to cisplatin-induced apoptosis, while the NRF2-knockdown A2780 cell line contains low ABCF2 levels and is more sensitive to cisplatin treatments. Furthermore, ABCF2 overexpression decreases apoptosis and increases cell viability following cisplatin treatment, indicating ABCF2 as a novel NRF2 target gene, playing a critical role in cisplatin resistance in ovarian cancer [84].

Solute carrier family 40 member 1 (SLC40A1) is an iron exporter, essential for iron metabolism homeostasis [85]. Interestingly, cisplatin-resistant ovarian cancer cells (A2780CP, COC1/DDP, PEO4) increase NRF2 levels and reduce SLC40A1 levels compared with cisplatin-sensitive cells (A2780, COC1, PEO1). Moreover, NRF2 knockdown leads to increased expression of SLC40A1, while NRF2 overexpression decreases SLC40A1 expression, showing that SLC40A1 expression is controlled by NRF2. In conclusion, SLC40A1 overexpression could reverse cisplatin resistance induced by NRF2, while SLC40A1 knockdown could restore cisplatin resistance and increase iron concentration. [86]. Studies discussed in this section are summarized in Table 3. 

## 5. NRF2 in Ovarian Function Preservation

Ovarian cancer is an important worldwide public health problem and thousands of young women are exposed to cytotoxic chemotherapy, causing DNA damage and apoptosis of oocytes, leading to infertility and ovarian aging [87,88]. Thus, preserving ovarian function and fertility in young women exposed to these therapies is important to guarantee a major quality-of-life factor in these patients. Currently, cryopreservation of gametes and ovarian tissues is an important tool to preserve fertility in young women, but it cannot reverse menopause or restore the ovarian function [89,90]. Thus, finding natural or synthetic compounds able to preserve ovarian function during chemotherapy may be a more useful way than cryopreservation, in maintaining fertility in these patients. In addition, cryopreservation of ovarian tissues induces ROS generation and oxidative stress that damage follicles, affecting the recovery efficiency and pregnancy rate.

Doxorubicin (DOX) is one of the most used antitumour drugs that causes gonad toxicity, inducing oxidative stress and leading to ovarian dysfunction [87]. Interestingly, Niringiyumukiza and colleagues found that DOX intraperitoneal injection in ICR mice increases NRF2, HO-1 and catalase (CAT), while reducing Glutathione peroxidase (GSH-Px) and SOD-1 expressions. DOX administration with SB216763, a potent Glycogen synthase kinase 3 (GSK-3) inhibitor, enhances NRF2 expression, restoring GSH-Px and SOD-1 expression levels. Moreover, DOX significantly decreases the number of primordial, primary, preantral and antral follicles, while it increases the number of atretic follicles, but these effects were reversed by SB216763 administration. Furthermore, SB216763 and DOX combined administration reduces the mature oocyte abnormalities, suggesting that GSK-3 and NRF2 crosstalk may reduce DOX-induced ovarian damages [91]. Thus, the use of naturals or synthetic GSK-3 inhibitors [92] could be useful tools for fertility preserving in young women undergoing DOX chemotherapy treatments.

Natural compounds (also called phytochemicals or phytonutrients) are biological substances present in plants (e.g., carotenoids, flavonoids, anthocyanins and polyphenols) that are normally used to protect themselves from external influences or against predators [93,94]. Although the precise mechanism of action of many natural compounds is unknown, these are normally used as worldwide supplements, showing beneficial effects in many diseases [93,95,96,97,98]. Despite remarkable toxicities, Busulfan, Cyclophosphamide and Melphan (Bu/Cy/Mel) is one of the most frequent drug combinations used in the treatment of patients with haematologic malignancy [99,100]. Interestingly, Wu and colleagues found that Resveratrol (3,5,4′-trihydroxy-trans-stilbene, RES), a natural phenol with antioxidant properties, derived from plants, relieves oogonial stem cells loss, attenuating the busulfan/cyclophosphamide (Bu/Cy)-induced oxidative apoptosis in mouse ovaries. RES could exert antioxidant function, activating NRF2, and it could be used combined to chemotherapeutics to prevent chemotherapy-induced ovarian aging [101].

Melatonin (N-acetyl-5-methoxytryptamine) is mainly synthesized and secreted by the pineal organ, acting as scavenger for many free radicals, and as an antioxidant, upregulating the expression of antioxidant proteins [102]. Interestingly, it has been shown that melatonin could protect follicular integrity, preventing cell apoptosis, decreasing ROS production, malondialdehyde (MDA) and nitric oxide (NO) levels. Moreover, melatonin could increase the activities of glutathione peroxidases (GSH-Px), GSH, CAT, and SOD in cryopreserved ovarian tissues. These effects may be related to NRF2 pathway activation, since the authors found increased NRF2 downstream genes, such as haem oxygenase-1 (HO-1), glutathione S-transferase M1 (GSTM1), SOD, and CAT. In summary, melatonin plays an important role in protecting follicular integrity during cryopreservation, acting not only as a direct ROS scavenger, but also inducing antioxidative enzymes activation, probably modulating the NRF2 pathway [103].

Cyclophosphamide (CTX) is a common drug used to treat female cancers but induces ROS production, damaging DNA and inducing follicular apoptosis, leading to early menopause and infertility [104]. For this reason, preserving female fertility during CTX treatment is very important. Interestingly, female mice exposed to CTX and treated with Epigallocatechin gallate (EGCG) and theaflavins (TFs), two natural compounds derived from green tea or black tea, improved the ovarian endocrine function and reproductivity. These natural compounds reduce DNA follicular damage due to oxidation induced by activating the NRF2/HO-1 and SOD2 pathways and reducing the apoptosis in growing follicles [105].

The organic solvent, 1-Bromopropane (1-BP), also known as n-propyl bromide, is widely used in industrial and commercial applications. However, it has been reported that exposure to 1-BP can cause toxic effects on the nervous system [106] and induce ovarian dysfunction [107]. Yang and colleagues showed that 1-BP treatments led to an increase in both ROS and MDA production and decreased SOD activity in OVCAR-3 cells. Moreover, they found that 1-BP activates NRF2, increases HO-1 expression and apoptosis. Interestingly, vitamin C alleviates 1-BP-induced apoptosis, activating the NRF2 pathway. Therefore, it can be deduced that 1-BP induces oxidative stress and apoptosis by inactivating NRF2 signalling in OVCAR-3 cells [108].

Cigarette smoke contains thousands of harmful components that have been reported to be dangerous on female reproductive organs, including ovaries, causing ovarian dysfunction. In fact, the use of Cigarette Smoke Extract (CSE), commonly used to simulate smoking effects in in vitro studies, showed that it can impair ovulation, oocyte morphology and causes apoptosis [109]. CSE treatments reduce cell proliferation by reducing Cyclins B1 and D1 expression, and induce apoptosis, modulating the Bcl-2 signalling in SKOV3 and OVCAR3 ovarian cancer cell lines. Additionally, CSE induces oxidative stress, increasing ROS levels and decreasing NRF2 expression by increasing KEAP1. This causes ovarian function damage, inducing the inhibition of cell proliferation and oxidative stress, probably by NRF2 activity [110]. 

## 6. Conclusions and Further Remarks

Recent studies show that the NRF2/KEAP1/ARE pathway plays a key role in many processes involved in the regulation of ovarian cancer progression, proliferation and chemoresistance. It is clear that NQO1 and NRF2 are highly expressed in ovarian carcinoma compared with normal tissues and that NRF2 expression increases with ovarian carcinoma stage advancing. Moreover, low KEAP1 expression is associated with disease recurrence and death, while high KEAP1 expression is predictive of better overall and disease-free survival. Interestingly, it has been reported that both low nuclear NRF2 expression and high KEAP1 expression depend on the age of patients, suggesting that efficiency in countering ROS decreases with aging and it is associated to an increased risk of carcinogenesis. Moreover, NRF2 can regulate ERα and PGR expressions in ovarian cancer cells, playing a pivotal role in cell response to oestrogen and progesterone (see Table 1 and Table 3).

Several lines of research have demonstrated that NRF2 signalling can be indirectly modulated by non-coding RNA, such as miR-181d and Lin-H19, and that these can modulate drug response in ovarian cancer [40,46]. Moreover, p62 can activate NRF2, protecting cancer cells from chemotherapeutic agents inducing autophagy [53,111]. Moreover, it is known a direct action of NRF2 on proteins involved in chemoresistance such as AKR1C1-3, ABCF2, SLC40A1, as well as on the modulation of important growth factor receptors, such as c-MET, ErbB2 and EGFR regulating tumour growth [76,81].

Unfortunately, chemotherapy causes cytotoxic effects, both in cancer and normal cells. Every year, thousands of young women are exposed to chemotherapy, with serious consequences on fertility and ovarian tissue [87]. To date, the ovarian tissues and gametes cryopreservation is the only way to give a chance to these women to become pregnant [89]. It has been reported that GSK-3 inhibitors, resveratrol, melatonin, epigallocatechin gallate, theaflavins can protect ovarian tissues exposed to chemotherapeutic agents or cryopreservation, activating NRF2 signalling to preserve female fertility (see Table 4).

In conclusion, NRF2 has many important functions, modulating different enzymes, receptors and miRNAs (see Figure 2). Thus, the NRF2/KEAP1/ARE pathway can protect cells from oxidative stress and regulate cancerous cells response to chemotherapeutic agents. Therefore, NRF2 can be considered a promising target for future research on ovarian cancer progression and treatment and could have a significant clinical impact in developing new therapies. Moreover, the use of natural or synthetic compounds activating NRF2 can play a pivotal role in ovarian function preservation in patients undergoing chemotherapy treatments.

## Figures and Tables

**Figure 1 antioxidants-11-00663-f001:**
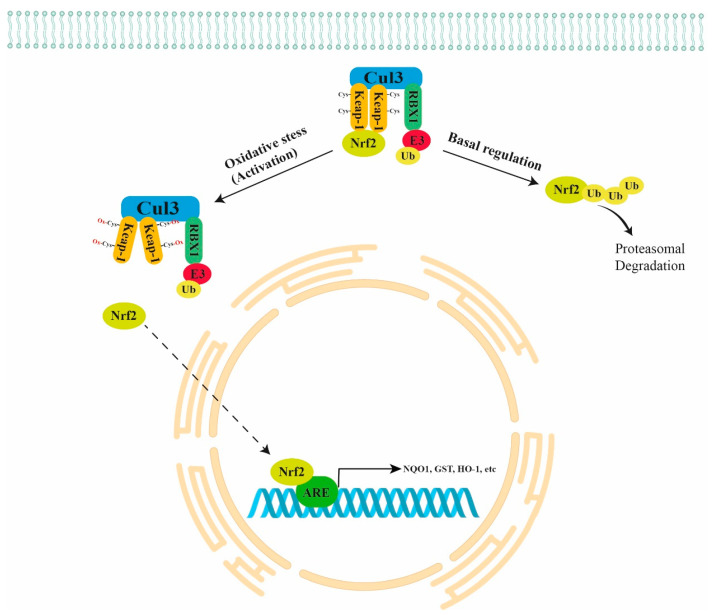
Schematic representation of NRF2 regulation. Normally, NRF2 is directly bound to the KEAP1/CUL3/RBX1 E3-ubiquitin ligase complex that targets NRF2 for proteasomal degradation. Under oxidant stimuli, ROS oxidate the cysteine residues of KEAP1 leading to a conformation change in KEAP1 that causes the inhibition of NRF2 ubiquitination and its translocation into the nucleus with consequent binding to the ARE regions of antioxidant genes (NQO1, GST, HO-1, etc.). KEAP1 = Kelch-like ECH Associated Protein 1; CUL3 = Cullin 3; RBX1 = RING-box protein 1.

**Figure 2 antioxidants-11-00663-f002:**
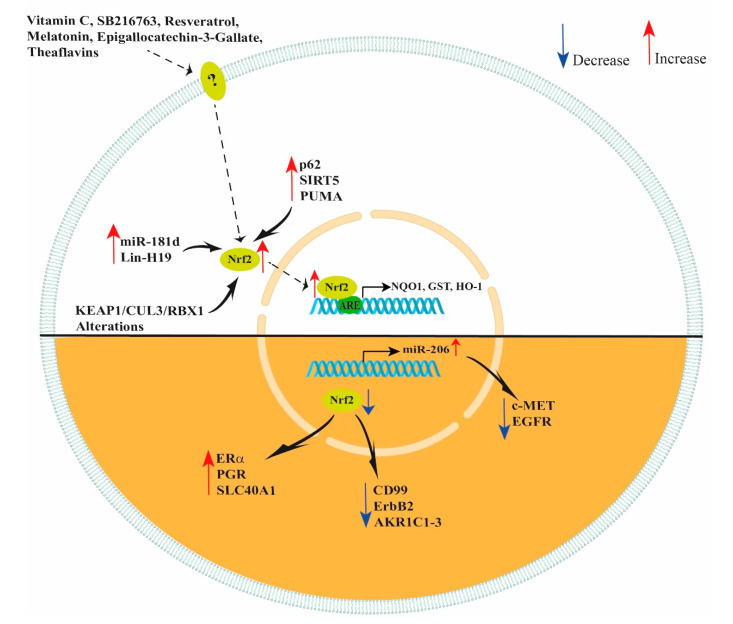
Schematic representation of NRF2 modulation. Vitamin C, SB216763, resveratrol, melatonin, epigallocatechin-3-gallate and theaflavins increase NRF2 expression in ovarian cells improving their response to oxidant agents. In ovarian cancer cells, increased levels of miR-181d, Lin-H19, p62, SIRT5, PUMA and KEAP1/CUL3/RBX1 alterations lead to an increase in NRF2 expression. However, decreased levels of NRF2 lead to an increased expression of ERα, PGR, SLC40A1, miR-206 and decreased expression of CD99, ErbB2 and AKR1C1-3. In addition, NRF2 indirectly decreases c-MET and EGFR expression by increasing miR-206 levels.

**Table 1 antioxidants-11-00663-t001:** NRF2 expression and correlation with ovarian cancer subtypes.

Tissues Studied	Proteins Analysed	Results	Ref.
10 normal tissues	NQO1 NRF2	NQO1 and NRF2 increased expression in ovarian carcinoma compared with normal and pre-cancerous lesions. NRF2 expression increases with ovarian carcinoma stage advancing	[25]
20 benign tumours:			
*12 serous*			
*8 mucinous*			
20 borderline tumours:			
*12 serous*			
*8 mucinous*			
50 ovarian carcinomas:			
*35 serous*			
*15 mucinous*			
108 ovarian carcinomas: *47 Serous* *23 Mucinous* *13 Endometrioid* *25 Clear cells*	KEAP1 E-cadherin NRF2	Serous carcinoma has a higher KEAP1 cytoplasmic, NRF2 nuclear expression and lower E-cadherin membrane positivity than mucinous, endometrioid and clear cell carcinomas. Patients with serous carcinoma are older in age and show highest KEAP1 expression and least percentage of E-cadherin immunoreactivity.	[26]
100 Clear cell carcinomas: *81 Chemosensitives* *19 Chemoresistants*	KEAP1 NRF2	Low KEAP1 expression is associated with disease recurrence and death. High KEAP1 expression is predictive of better overall and disease-free survival. No association among chemoresistance, NRF2 and KEAP1 expression is detected but patients with high KEAP1 expression have significantly lower recurrence rates and death. Significant and positive correlations between the intensities of cytoplasmic NRF2 and KEAP1 expression.	[27]
156 EOC: *110 serous* *21 endometrioid* *12 clear cells* *13 mucinoses*	ERα NRF2	Nuclear NRF2 expression is low in serous, clear cell, and endometrioid carcinomas but high in the mucinous subtype. Low nuclear NRF2 expression is associated with age of patients. No association of ERα expression among subtypes but high ERα expression is present in low-graded carcinomas compared to high-graded ones. NRF2 cytoplasmic expression correlates with ERα expression. Both NRF2 cytoplasmic and ERα expressions are associated with longer overall survival in serous carcinoma.	[28]
156 EOC: *110 serous* *21 endometrioid* *12 clear cells* *13 mucinoses*	PRA PRB NRF2	NRF2 cytoplasmic expression is correlated with both PRA and PRB expressions, and is associated with a significant impact on overall survival. Grading, FIGO, lymph node involvement (pN), and distant metastasis (pM) show no significant differences.	[29]

EOC: Epithelial ovarian cancer; PRA: Progesterone Receptor A; PRB: Progesterone Receptor B; ERα: Estrogen Receptor α.

**Table 2 antioxidants-11-00663-t002:** NRF2 cellular modulators in ovarian cancer.

Modulator	Model Studied	Results	Ref.
miR-181d	Ovarian tissues A2780 and A2780/DDP cells	Increased miR-181d expression in ovarian tissues of DDP-resistant patients and in the A2780/DDP cell line. MiR-181d increases DDP resistance by downregulating OGT that represses NRF2 expression through glycosylation of KEAP1.	[40]
Lin-H19	A2780 and A2780/DDP cells	Increased expression of LIN-RECK-3, H19, LUCAT1, LINC00961 and linc-CARS2-2 in A2780/CDDP cells. Lin-H19 knockdown in A2780/CDDP cells leads to cisplatin sensitivity and reduces the expression of NQO1, GSR, G6PD, GCLC, GCLM and GSTP1.	[46]
p62/SQSTM1	SKOV3 and SKOV3/CDDP cells	SKOV3/CDDP has higher levels of p62 than the cisplatin-sensitive SKOV3 cells. P62 activates KEAP1-NRF2-ARE pathway that induces the expression of antioxidant genes in SKOV3/DDP cells.	[53]
KEAP1/CUL3/RBX1 E3-ubiquitin ligase complex alterations	Serous ovarian carcinomas (OVCA) patients	Almost 90% of OVCA cases shows function alterations in any components of the NRF2 inhibitory complex. Copy-number loss (CNL) is the most prominent disruption mechanism and most frequently observed in RBX1 component. High frequency of DNA and mRNA alterations of the KEAP1/CUL3/RBX1 complex leads to high levels of NRF2 in OVCA.	[60]
SIRT5	SKOV-3, CAOV-3 and A2780 cells	SIRT5 levels are higher in cisplatin-resistant SKOV-3 and CAOV-3 ovarian cancer cells than in cisplatin-sensitive A2780 cells. Overexpression of SIRT5 in A2780 cells facilitates cell growth and cisplatin-resistance suppressing cisplatin-induced DNA damage by increasing NRF2 and HO-1 expression.	[56]
KEAP1 mutations	Epithelial ovarian cancer (EOC) patient specimens	Nuclear NRF2 is present in over half of EOC specimens with a more frequency in clear cell subtype and upregulation of NRF2 target genes. Genetic mutations of KEAP1 sequence in 29% of clear cell carcinoma samples and 8% of other subtypes. Patients with active NRF2 pathway show resistance to platinum-based therapy and lower overall survival.	[58]
PUMA	A2780 and SKOV3 cells	PUMA-overexpressed in A2780 and SKOV3 cells shows increased ROS generation and increased NRF2, HO-1 expression and apoptosis.	[64]

**Table 3 antioxidants-11-00663-t003:** NRF2 cellular targets in ovarian cancer.

Protein Regulated by NRF2	Model Studied	Effect	Ref.
ERα	OVCAR3, ES2, UWB1.289, and TOV112D ovarian cancer cells and HOSEpiC (benign cells)	NRF2 silencing increases ESR1 expression in OVCAR3 and ES2 cells. NRF2 is highly expressed in the ovarian cancer cell lines OVCAR3, ES2, UWB1.289, and TOV112D compared with the benign cell line HOSEpiC. ERα, is reduced in all ovarian cancer cell lines compared to the benign cell line HOSEpiC.	[28]
CD99	A2780, A2780/CDDP, COC1 and COC1/CDDP cells	CD99 is highly expressed in cisplatin-resistant both ovarian cancer cells (A2780/CDDP and COC1/CDDP) and tissues compared to both ovarian cisplatin-sensitive cells (A2780 and COC1) and tissues. CD99 overexpression leads to cisplatin resistance while CD99 knockdown sensitizes ovarian cancer cells to cisplatin. NRF2 silencing leads to decreased CD99 expression and cell viability after cisplatin treatment in cisplatin-resistant cells.	[74]
ErbB2	SKOV3 cells	NRF2 silencing represses NRF2 signaling leading to cell growth G0/G1 phase arrest, tumour growth retardation and a significant decrease of ErbB2 expression in mouse xenografts. ErbB2 downregulation leads to a decrease in pAKT and increase p27 protein, enhancing the effect of NRF2 knockdown in SKOV3 growth.	[76]
AKR1C1 AKR1C2 AKR1C3	SKOV3 cells	NRF2 knockdown decreases AKR1C1-3 expression and increases ROS production after cisplatin treatment. Moreover, NRF2 knockdown increases activation of the pJNK/p38 pathway and decreases phosphorylation of ATF2.	[79]
c-MET EGFR	SKOV3 cells	NRF2 silencing increases miR-206 expression and reduces the levels of c-MET and EGFR inhibiting cell proliferation and increasing doxorubicin effect in SKOV3 cells.	[81]
PGR	OVCAR3, ES2, UWB1.289, HOSEpiC and TOV112D cells	NRF2 is increased and PGR decreased in the ovarian cancer cell lines compared with the benign line (HOSEpiC). NRF2 silencing induces higher PGR mRNA expression in OVCAR3.	[29]
ABCF2	A2780 cells	ABCF2 has a functional antioxidant response element (ARE) in its promoter region that is regulated by NRF2 responsible for cisplatin resistance.	[84]
SLC40A1	cisplatin-sensitive (A2780, COC1, PEO1) and cisplatin-resistant (A2780CP, COC1/DDP, PEO4) cells	Increased levels of NRF2 and reduced levels of SLC40A1 in cisplatin-resistant cells compared with cisplatin-sensitive cells. NRF2 knockdown leads to SLC40A1 overexpression while NRF2overexpression caused SLC40A1 downregulation. SLC40A1 overexpression reverses cisplatin resistance induced by NRF2, while SLC40A1knockdown restores cisplatin resistance and increases iron concentration.	[86]

**Table 4 antioxidants-11-00663-t004:** Modulators of NRF2 in ovarian preservation.

Protector Compound	Harmful Agent	Mode Studied	Effect	Ref.
SB216763 (GSK-3 inhibitor)	Doxorubicin (DOX)	Mice	SB216763 and DOX combined treatment enhances NRF2 expression restoring GSH-Px and SOD-1 levels. SB216763 increases primordial, primary, preantral and antral follicles number while decreases atretic follicles number. SB216763 and DOX coadministration reduces the mature oocyte abnormalities.	[91]
Resveratrol (RES)	Busulfan and Cyclophosphamide (Bu/Cy)	Mice	RES activates NRF2 and relieves oogonial stem cells loss attenuating the Bu/Cy-induced oxidative apoptosis in mouse ovaries.	[101]
Melatonin	ROS due to cryopreservation	Rat ovarian tissues	Melatonin increases GSH-Px, GSH, CAT and SOD activities in cryopreserved ovarian tissues by activating NRF2 downstream genes HO-1, GSTM1, SOD, and CAT.	[103]
Epigallocatechin gallate (EGCG) Theaflavins (TFs)	Cyclophosphamide (CTX)	Mice	Mice exposed to CTX and treated with EGCG and TFs improve ovarian endocrine function and reproductivity reducing the oxidation-induced follicular DNA damage by activating the NRF2/HO-1 and SOD2 pathways and reducing the apoptosis of growing follicles.	[105]
Vitamin C	1-BromoPropane (1-BP)	OVCAR-3 cells	1-BP treatment leads to increased ROS and MDA production and decreased SOD activity. Vitamin C alleviates 1-BP-induced apoptosis activating NRF2 pathway.	[108]
—	Cigarette Smoke Extract (CSE)	SKOV3 and OVCAR3 cells	CSE reduces cell proliferation by Cyclins B1 and decreases D1 expression, and induces apoptosis. CSE induces oxidative stress increasing ROS levels and repressing NRF2 expression by increasing KEAP1.	[110]

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
