# Peer review of "Role of NRF2 in Ovarian Cancer"

_antioxidants, 2022, doi:10.3390/antiox11040663_

Round 1

Reviewer 1 Report

Nicely written review.

I have only few comments. In the legend of Figure 1 could you please explain the abbreviations for KEAP1/CUL3/RBX1? I have seen that it is explained in the text but for any reader it makes it better to understand.

In addtion, it would be nice if the authors could include a summarizing figure as figure 2.

Could you please consider to include a graphical abstract?

Author Response

Reviewer 1

I have only few comments. In the legend of Figure 1 could you please explain the abbreviations for KEAP1/CUL3/RBX1? I have seen that it is explained in the text but for any reader it makes it better to understand.

We apologise with the reviewer, we added the full name of the abbreviations in the figure legend.

In addition, it would be nice if the authors could include a summarizing figure as figure 2.

We made a summarizing figure as requested by the reviewer (figure 2)

Could you please consider to include a graphical abstract?

We added a graphical abstract as requested by the reviewer

Reviewer 2 Report

As the authors describe in their review, the role of Nrf2 in ovarian cancer is complicated and context-dependent. Thus, the review is a useful summary of the current state of the field. Notably, the section on ovarian function preservation is especially valuable as this important topic is rarely described in the Nrf2 literature. Overall, this review will be of interest to a wide audience.

Minor suggestions:

The organization of the review could be improved. Adding headings to separate topics or additional transition sentences would help the reader. Single or two sentence paragraphs (line 90, 185, 209, 412, 416) should be avoided.

Grammatical errors found throughout the manuscript should be corrected.

Many of the studies summarized in Table 1 reach contradictory conclusions. What factors might contribute to these disparate results and how should the reader interpret these studies?

The tables should be reformatted with less text in the results column.

Author Response

Reviewer 2

The organization of the review could be improved. Adding headings to separate topics or additional transition sentences would help the reader. Single or two sentence paragraphs (line 90, 185, 209, 412, 416) should be avoided.

We thank the reviewer for the suggestion. We modified the paragraphs suggested.

Grammatical errors found throughout the manuscript should be corrected.

Text has been deeply checked and grammar errors throughout the manuscript has been corrected

Many of the studies summarized in Table 1 reach contradictory conclusions. What factors might contribute to these disparate results and how should the reader interpret these studies?

We discussed the reasons of these contradictory conclusions in lines 156-158.

The tables should be reformatted with less text in the results column.

We thank the reviewer, we already tried to reduce the text in the results column but there could be the risk to eliminate important information. However, we tried to reduce the text in all tables (when possible).